# Mangrove Tirucallane- and Apotirucallane-Type Triterpenoids: Structure Diversity of the C-17 Side-Chain and Natural Agonists of Human Farnesoid/Pregnane–X–Receptor

**DOI:** 10.3390/md16120488

**Published:** 2018-12-06

**Authors:** Zhong-Ping Jiang, Zhi-Lin Luan, Ruo-Xi Liu, Qun Zhang, Xiao-Chi Ma, Li Shen, Jun Wu

**Affiliations:** 1Marine Drugs Research Center, College of Pharmacy, Jinan University, 601 Huangpu Avenue West, Guangzhou 510632, China; jiangzhongping0113@163.com (Z.-P.J.); 13249133788@163.com (R.-X.L.); zhangqun0917@163.com (Q.Z.); 2College of Pharmacy and Advanced Institute for Medical Sciences, Dalian Medical University, Dalian 116044, China; zhilin_luan@sina.com (Z.-L.L.); maxc1978@163.com (X.-C.M.); 3School of Pharmaceutical Sciences, Southern Medical University, 1838 Guangzhou Avenue North, Guangzhou 510515, China

**Keywords:** mangrove, triterpenoid, tirucallane, apotirucallane, farnesoid–X–receptor, pregnane–X–receptor

## Abstract

Ten new triterpenoid compounds with structure diversity of the C-17 side-chain, including nine tirucallanes, named xylocarpols A–E (**1**–**5**) and agallochols A–D (**6**–**9**), and an apotirucallane, named 25-dehydroxy protoxylogranatin B (**10**), were isolated from the mangrove plants *Xylocarpus granatum*, *Xylocarpus moluccensis*, and *Excoecaria agallocha*. The structures of these compounds were established by HR-ESIMS and extensive one-dimensional (1D) and two-dimensional (2D) NMR investigations. The absolute configurations of **1** and **2** were unequivocally determined by single-crystal X-ray diffraction analyses, conducted with Cu K*α* radiation; whereas those of **4**, **6**–**8** were assigned by a modified Mosher’s method and the comparison of experimental electronic circular dichroism (ECD) spectra. Most notably, **5**, **6**, **7**, and **9** displayed potent activation effects on farnesoid–X–receptor (FXR) at the concentration of 10.0 μM; **10** exhibited very significant agonistic effects on pregnane–X–receptor (PXR) at the concentration of 10.0 nM.

## 1. Introduction

Cholestasis, a clinical syndrome of hepatobiliary diseases, is usually caused by accumulation of bile acids in the liver and systemic circulation [1]. Long-term cholestasis can lead to primary biliary cirrhosis, primary sclerosing cholangitis, and hepatic failure. In clinical practice, abnormal metabolism of bile acids is deemed to be a crucial risk factor that induces cholestasis and cholestatic liver injury [1,2]. Farnesoid–X–receptor (FXR) and pregnane–X–receptor (PXR) are two members of the nuclear receptor family. Due to the regulation capability of a suite of genes involved in the metabolism, transport, and elimination of bile acids, FXR and PXR are considered to be the key target proteins for the treatment of cholestasis and liver injury [1,3,4,5].

*Xylocarpus granatum* and *Xylocarpus moluccensis*, true-mangrove species of the family Meliaceae, have been used in folk medicine, particularly in South and Southeast Asian countries, for the treatment of cholera, diarrhea, malaria, and other fever diseases [6]. Mangroves of the genus *Xylocarpus* are mainly distributed in East Africa, India, Bangladesh, Southeast Asia, Southern China, and Northern Australia [7,8]. Previous chemical investigation of *X. granatum* and *X. moluccensis*, collected from different locations, led to the isolation and identification of various limonoids, protolimonoids, alkaloids, and flavanones. To the best of our knowledge, limonoids are the main secondary metabolites of these mangroves [7,8,9].

*Excoecaria agallocha*, a semi-mangrove species of the family Euphorbiaceae, has been used in Asia as a traditional medicine for the treatment of epilepsy, haematuria, leprosy, and toothache [10,11]. It is mainly distributed along the sea coasts of China, India, Philippines, and Oceania. The poisonous milky latex, exuded from the bark of this plant, can cause blistering of the skin and temporary blindness [12,13,14,15,16]. Previous chemical investigation of this plant resulted in the isolation and identification of diterpenoid, triterpenoid, tanninoid, and flavonoid natural compounds [10,17,18,19,20,21]. To date, only 18 triterpenoid compounds, including types of cycloartane, friedelane, lupane, oleanane, and taraxerane, have been reported from this plant [17].

As part of our ongoing investigation of bioactive natural products from mangrove plants, 10 new triterpenoid compounds, including nine tirucallanes (**1**–**9**) and one apotirucallane (**10**) (Figure 1), were isolated from mangrove plants *X*. *granatum*, *X*. *moluccensis*, and *E*. *agallocha*. We reported the isolation and structure elucidation of these compounds, and their agonistic effects on human FXR and PXR.

## 2. Results and Discussion

Compound **1** was obtained as colorless crystals. Its molecular formula C_30_H_50_O_5_ with six indices of hydrogen deficiency was established by the negative HR-ESIMS quasi molecular ion peak at *m*/*z* 525.3356 ([M + Cl]^−^, calculated for 525.3352) (Appendix A). According to the ^1^H and ^13^C NMR spectroscopic data of **1** (Table 1 and Table 2) (Appendix A), two elements of unsaturation were due to a carbon-carbon double bond and a carbonyl group. Thus, the molecule was tetracyclic. The ^13^C NMR spectroscopic data and the DEPT135 experiment of **1** (Appendix A) revealed the presence of eight methyl groups, seven methylene groups, eight methine groups (including three oxygenated ones at *δ*_C_ 75.8, 70.9, and 80.1 ppm, respectively, and an olefinic one at *δ*_C_ 118.0 ppm), and seven quaternary carbons (including an oxygenated one at *δ*_C_ 74.1, an olefinic one at *δ*_C_ 145.7, and a ketone carbon at *δ*_C_ 217.0 ppm, respectively).

^1^H‒^1^H COSY correlations (Appendix A) between H-22/H-23 and H-23/H-24 and HMBC correlations from H-24 to C-25, C-26, and C-27 confirmed the connection of the fragment of C-22(OH)–C-23(OH)–C-24(OH)–C-25(OH)–(Me)_2_ (Figure 2a). The NMR spectroscopic data of **1** resembled those of toonaciliatavarin E [22], except for different orientations of three chiral centers on the C-17 side-chain, videlicet C-22, C-23, and C-24.

The relative configuration of the tetracyclic tirucallane core (rings-A, B, C, and D) of **1** was corroborated by diagnostic NOE interactions (Appendix A). Those between H-5/H-9, H-5/H_3_-29, H-9/H_3_-18, and H_3_-18/H-20 revealed the *α*-oriented H-5, H-9, H_3_-18, H-20, and H_3_-29. Similarly, NOE interactions between H_3_-19/H-2*β*, H_3_-19/H_3_-28, H_3_-19/H-11*β*, H_3_-30/H-11*β*, and H_3_-30/H-17 assigned the *β*-oriented H-17, H_3_-19, H_3_-28, and H_3_-30 (Figure 2b).

In order to establish the absolute configuration of the whole molecule of **1**, particularly the absolute configuration of three chiral centers on the C-17 side-chain, a single-crystal X-ray diffraction analysis, conducted with Cu K*α* radiation (Flack parameter of 0.01(6)) (Figure 3), was employed. Thus, the absolute configuration of **1**, named xylocarpol A, was unequivocally assigned as (5*R*,9*R*,10*R*,13*S*,14*S*,17*S*,20*R*,22*S*,23*S*,24*S*)-22,23,24,25-tetrahydroxytirucalla-7-ene-3-one.

Compound **2** was isolated as colorless crystals. Its molecular formula C_30_H_48_O_3_ (seven degrees of unsaturation) was established by the positive HR-ESIMS quasi molecular ion peak at *m*/*z* 457.3670 ([M + H]^+^, calculated for 457.3676) (Appendix A). The NMR spectroscopic data of **2** (Table 1 and Table 2) (Appendix A) were closely related to those of aphagranin D [23], the difference being the absence of the 24-OH and 25-OH groups in **2**. This deduction were corroborated by the upshifted C-24 (*δ*_C_ 82.1 CH in aphagranin D; whereas *δ*_C_ 46.8 CH_2_ in **2**) and C-25 (*δ*_C_ 79.0 qC in aphagranin D; whereas *δ*_C_ 24.5 CH in **2**), and the proton spin-spin system, i.e., H_2_-24‒H-25(H_3_-26)‒H_3_-27, observed in the ^1^H‒^1^H COSY spectrum of **2** (Appendix A).

NOE interactions (Appendix A) between H-5/H-9, H-5/H_3_-29, H-9/H_3_-18, and H_3_-18/H-20 assigned the *α*-oriented H-5, H-9, H_3_-18, and H_3_-29; whereas those between H_3_-19/H-2*β*, H_3_-19/H_3_-28, H_3_-30/H-12*β*, H_3_-30/H-17 concluded the *β*-oriented H-17, H_3_-19, H_3_-28, and H_3_-30. Finally, single-crystal X-ray diffraction analysis, conducted with Cu K*α* radiation (Flack parameter of 0.08(13)) (Figure 4), unambiguously established the absolute configuration of the whole molecule of **2**. Therefore, the absolute configuration of **2**, named xylocarpol B, was unambiguously determined to be (5*R*,9*R*,10*R*,13*S*,14*S*,17*S*,20*R*,22*S*)-22-hydroxytirucalla-7-ene-3,23-dione.

Compound **3** provided the molecular formula C_30_H_48_O_3_ as deduced from the positive HR-ESIMS quasi molecular ion peak at *m*/*z* 457.3654 ([M + H]^+^, calculated for 457.3676) (Appendix A). The NMR spectroscopic data of **3** (Table 1 and Table 2) (Appendix A) resembled those of **2**, except for the different location of the only hydroxy group on the C-17 side-chain. A 24-OH group was observed in **3** instead of the 22-OH function in **2**. HMBC correlations (Appendix A) from the oxygenated H-24 (*δ*_H_ 4.07, s) to C-23, C-25, C-26, and C-27 confirmed the presence of the 24-OH group. NOE interactions (Appendix A) between H-5/H-9, H-5/H_3_-29, H-9/H_3_-18, and H_3_-18/H-20, and those between H_3_-19/H-2*β*, H_3_-19/H_3_-28, H_3_-30/H-11*β*, and H_3_-30/H-17 established the same tetracyclic tirucallane core (rings-A, B, C, and D) in **3** as that of **2**. However, due to the limited amount of **3**, the chirality of C-24 could not be determined. Thus, the structure of **3**, named xylocarpol C, was assigned as 24-hydroxytirucalla-7-ene-3,23-dione.

Compound **4** gave the molecular formula C_30_H_50_O_4_ as obtained from the negative HR-ESIMS quasi molecular ion peak at *m*/*z* 509.3402 ([M + Cl]^−^, calculated for 509.3403) (Appendix A). The ^1^H and ^13^C NMR spectroscopic data of **4** (Table 1 and Table 2) (Appendix A) were closely related to those of odoratone [24], except for the replacement of the C-3 ketone group in odoratone by a hydroxy group in **4**. The above deduction was corroborated by the upshifted C-3 (*δ*_C_ 216.0 qC in odoratone; whereas *δ*_C_ 79.3 CH in **4**) and ^1^H‒^1^H COSY correlations between H_2_-2/H-3 (Figure 5a) (Appendix A). The relative configuration of **4** was established by NOE interactions (Appendix A). Those between H-3/H-5, H-5/H-9, H-5/H_3_-29, H-9/H_3_-18, and H_3_-18/H-20 assigned the *α*-oriented H-3, H-5, H-9, H_3_-18, H-20, and H_3_-29; NOE interactions between H_3_-19/H_3_-28, H_3_-30/H-11*β*, H_3_-19/H-11*β*, and H_3_-30/H-17 concluded the *β*-oriented H-17, H_3_-19, H_3_-28, and H_3_-30 (Figure 5b). The above results concluded the same relative configuration for the tetracyclic tirucallane core (rings-A, B, C, and D) in **4** as that of **1**–**3**, except for the additional chiral C-3 in **4**. Moreover, NOE interactions between H-23/H_3_-21 and H-24/H_3_-26 assigned the *α*-oriented H-23, H-24, and H_3_-26; those between H-22/H_3_-27 concluded the *β*-oriented H-22 and H_3_-27 (Figure 5b). Thus, the relative configuration of the C-17 side-chain of **4** was determined.

In order to establish the absolute configuration of **4**, a modified Mosher’s *α*-methoxy-*α*-(trifluoromethyl)phenylacetyl (MTPA) ester method was applied [25]. The (3,23,24)-tri(*S*)- and (3,23,24)-tri(*R*)-MTPA esters of **4**, videlicet **4s** and **4r** (Appendix A), were successfully prepared. Based on the MTPA ester rule of Δ*δ* (*δ_S_* − *δ_R_*) values (Figure 6) [25], the absolute configurations of C-3, C-23, and C-24 were assigned as *S*, *R*, and *S*, respectively. Therefore, the absolute configuration of **4**, named xylocarpol D, was unequivocally established as (3*S*,5*R*,9*R*,10*R*,13*S*,14*S*,17*S*,20*R*,22*S*,23*R*,24*S*)- 3,23,24-trihydroxy-22,25-epoxytirucalla-7-ene. The absolute configuration of the 2,2-dimethyltetrahydrofuran-3,4-diol moiety of odoratone [24] was first clarified as 22*S*,23*R*,24*S*.

The molecular formula of **5** was determined to be C_30_H_48_O_4_ (seven degrees of unsaturation) by the negative HR-ESIMS quasi molecular ion peak at *m*/*z* 507.3245 ([M + Cl]^−^, calculated for 507.3247) (Appendix A). Two elements of unsaturation were due to a carbon-carbon double bond and a ketone group; thus, the molecule was pentacyclic. The NMR spectroscopic data of **5** (Table 1 and Table 2) (Appendix A) were similar to those of **4**, the difference being the opposite orientation of the 24-OH group and the replacement of the 3-OH group in **4** by a ketone function in **5**. Diagnostic NOE interactions (Appendix A) between H-22/H_3_-27 and H-24/H_3_-27 assigned the *β*-oriented H-22 and H-24; whereas those between H-23/H_3_-26 concluded the *α*-oriented H-23 (Figure 7). HMBC correlations (Appendix A) from H_2_-1, H_2_-2, H_3_-28, and H_3_-29 to the carbon (*δ*_C_ 217.2 qC) of a ketone group confirmed its location at C-3. Therefore, the structure of **5**, named xylocarpol E, was assigned as depicted.

The molecular formula of **6** was concluded to be C_30_H_50_O_4_ by the positive HR-ESIMS quasi molecular ion peak at *m*/*z* 475.3790 ([M + H]^+^, calculated for 475.3782) (Appendix A). The ^1^H and ^13^C NMR spectroscopic data of **6** (Table 3 and Table 4) (Appendix A) resembled those of brumollisol B [26], except for the absence of the 23-OH group, being corroborated by the upshifted C-23 (*δ*_C_ 69.5 CH in brumollisol B; whereas *δ*_C_ 28.4 CH_2_ in **6**), ^1^H‒^1^H COSY correlations (Appendix A) between H_2_-23/H-24, and HMBC correlations (Appendix A) from H_3_-26 and H_3_-27 to C-24 (Figure 8a).

The relative configuration for the tetracyclic tirucallane core (rings-A, B, C, and D) of **6** was established by NOE interactions (Appendix A). Those between H-3/H-5, H-5/H-9, H-5/H_3_-29, H-9/H_3_-18, and H_3_-18/H-20 assigned the *α*-oriented H-3, H-5, H-9, H_3_-18, H-20, and H_3_-29; NOE interactions between H_3_-19/H_3_-28, H_3_-30/H-11*β*, H_3_-19/H-11*β*, and H_3_-30/H-17 concluded the *β*-oriented H-17, H_3_-19, H_3_-28, and H_3_-30 (Figure 8b).

The absolute configuration of **6** was established by the application of the modified Mosher’s MTPA ester method [25]. The (3,24)-di(*S*)- and (3,24)-di(*R*)-MTPA esters of **6** were prepared (Appendix A). Based on the MTPA ester rule of Δ*δ* (*δ_S_* − *δ_R_*) values (Figure 9) [25], the absolute configurations of C-3 and C-24 were assigned as *S* and *S*, respectively. Hence, the absolute configuration of **6**, named agallochol A, was unequivocally assigned as (3*S*,5*R*,9*R*,10*R*,13*S*,14*S*,17*S*,20*S*,24*S*)- 3,24,25-trihydroxytirucalla-7-ene-6-one.

Compound **7** provided the same molecular formula (C_30_H_50_O_4_) as that of **6** based on the positive HR-ESIMS quasi molecular ion peak at *m*/*z* 475.3792 ([M + H]^+^, calculated for 475.3782) (Appendix A). The ^1^H and ^13^C NMR spectroscopic data of **7** (Table 3 and Table 4) (Appendix A) resembled those of **6**, except for the slight difference of ^1^H and ^13^C chemical shifts of CH-24 (*δ*_H_ 3.36 (br s), *δ*_C_ 78.6 in **6**; whereas *δ*_H_ 3.30 dd (*J* = 11.6, 1.6 Hz), *δ*_C_ 79.5 in **7**), indicating that both compounds are a pair of C-24 epimers. The above deduction was further corroborated by ^1^H‒^1^H COSY correlations between H-24/H_2_-23 (Appendix A) and HMBC interactions between H-24/C-23, H-24/C-25, H_3_-26/C-24, and H_3_-27/C-24 (Appendix A).

In order to determine the absolute configurations of C-3 and C-24 of **7**, (3,24)-di(*S*)- and (3,24)-di(*R*)-MTPA esters of **7** were prepared (Appendix A). Based on the MTPA ester rule of Δ*δ* (*δ_S_* − *δ_R_*) values (Figure 9) [25], the absolute configurations of C-3 and C-24 were assigned as *S* and *R*, respectively. Hence, the absolute configuration of **7**, named agallochol B, was unambiguously concluded to be (3*S*,5*R*,9*R*,10*R*,13*S*,14*S*,17*S*,20*S*,24*R*)-3,24,25-trihydroxytirucalla-7-ene-6-one.

The molecular formula of **8** was established as C_30_H_48_O_3_ by the positive HR-ESIMS quasi molecular ion peak at *m*/*z* 457.3673 ([M + H]^+^, calculated for 457.3682) (Appendix A). The NMR spectroscopic data of **8** (Table 3 and Table 4) (Appendix A) were similar to those of **7**, except for the presence of a Δ^23(24)^ double bond (*δ*_H_ 5.60 (dd, *J* = 15.6, 4.8 Hz, 1H), 5.62 (d, *J* = 15.6 Hz, 1H); *δ*_C_ 125.2, CH, 139.7, CH) and the absence of the 24-OH group. ^1^H‒^1^H COSY correlations between H-23/H_2_-22 and H-23/H-24 (Appendix A), and HMBC correlations between H-23/C-22, H-23/C-24, H-23/C-25, H-24/C-22, H-24/C-23, and H-24/C-25 in **8** (Appendix A) confirmed the above deduction. In addition, the coupling constant of 15.6 Hz between H-23 and H-24 assigned the *E*-geometry of the Δ^23(24)^ double bond.

The relative configuration of the tetracyclic tirucallane core (rings-A, B, C, and D) of **8** was established as the same as that of **7** by NOE interactions (Appendix A). Those between H-5/H-3, H-5/H-9, H-5/H_3_-29, H-9/H_3_-18, H_3_-18/H-20, H_3_-19/H_3_-28, H_3_-19/H_3_-30, and H_3_-30/H-17 assigned the same relative configuration of **8** as that of **7**. Moreover, the absolute configuration of **8**, particularly that of the tetracyclic tirucallane core (rings-A, B, C, and D), was established to be the same as that of **7**, except for the deficiency of the chiral C-24, by the accurate fit of their experimental ECD spectra (Figure 10). Therefore, the structure of **8**, named agallochol C, was determined to be (3*S*,5*R*,9*R*,10*R*,13*S*,14*S*,17*S*,20*S*)-3,25-dihydroxytirucalla-7,23-diene-6-one.

Compound **9** had the molecular formula C_27_H_44_O_3_ as determined from its negative HR-ESIMS quasi molecular ion peak at *m*/*z* 451.2985 ([M + Cl]^−^, calculated for 451.2984) (Appendix A). The NMR spectroscopic data of **9** (Table 3 and Table 4) (Appendix A) resembled those of a trinortirucalla-7-ene, i.e., sikkimenoid F [27], except for the replacement of the C-24 aldehyde group in sikkimenoid F by a C-24 carboxyl group in **9**. The above deduction was corroborated by the upshifted C-24 (*δ*_H_ 9.75 (br s), *δ*_C_ 203.1 CH in sikkimenoid F; whereas *δ*_C_ 178.5 qC in **9**) and HMBC correlations from H_2_-22 and H_2_-23 to the carbonyl carbon (C-24) of this carboxyl group. The relative configuration of the tetracyclic tirucallane core (rings-A, B, C, and D) of **9** was established by NOE interactions (Appendix A). Those between H-5/H-3, H-5/H-9, H-5/H_3_-29, H-9/H_3_-18, H_3_-18/H-20, H_3_-19/H_3_-28, H_3_-19/H_3_-30, and H_3_-30/H-17 assigned the same relative configuration of the tetracyclic tirucallane core of **9** as that of **8**. Thus, the structure of **9**, named agallochol D, was assigned as 3*β*-hydroxy-25,26,27-trinortirucalla-7-ene-24-oic acid.

Compound **10** provided the molecular formula C_32_H_46_O_7_ based on the positive HR-ESIMS quasi molecular ion peak at *m*/*z* 543.3303 ([M + H]^+^, calculated for 543.3316) (Appendix A), requiring ten degrees of unsaturation. According to the ^1^H and ^13^C NMR spectroscopic data of **10** (Table 3 and Table 4) (Appendix A), six of the 10 elements of unsaturation were due to two carbon-carbon double bonds and four carbonyls. Thus, the molecule was tetracyclic. The DEPT135 experiment of **10** (Appendix A) combined with its ^13^C NMR spectroscopic data revealed the presence of eight methyl groups, six methylene groups, nine methine groups (including three olefinic ones), and nine quaternary carbons (including four carbonyl carbons).

The above NMR characteristic features of **10** closely resembled those of an apotirucallane protolimonoid, i.e., protoxylogranatin B [28], except for the absence of the 25-OH group in **10**. ^1^H–^1^H COSY correlations from the proton of a methine moiety (*δ*_H_ 2.56 m; *δ*_C_ 34.1) to H_3_-26 and H_3_-27 confirmed the presence of the CH-25 group (Figure 11a). The relative configuration of **10** was established on the basis of NOE interactions (Appendix A). Those between H-5/H-9, H-5/H_3_-29, H-9/H_3_-18, H-20/H_3_-18 assigned the *α*-oriented H-5, H-9, H_3_-18, and H-20; whereas those between H_3_-19/H_3_-28, H_3_-19/H-6*β*, and H_3_-30/H-6*β* concluded the *β*-oriented H_3_-19 and H_3_-30 (Figure 11b). The NOE interaction between H-7/H_3_-30, but not between H-7/H-5 and H-7/H_3_-18, indicated the *β*-oriented H-7 and the corresponding *α*-oriented 7-acetoxy group (Figure 11b). Thus, the structure of **10**, named 25-dehydroxy protoxylogranatin B, was established as depicted.

In order to search for natural agonists of human FXR and PXR, most of the above isolated compounds were screened for their agonistic effects on these nuclear receptors. Chenodeoxycholic acid (CDCA) or rifampicin was used as the positive control at the concentration of 80.0 μM or 10.0 μM, respectively (Figure 12 and Figure 13). The results showed that **6** and **7** displayed significant agonistic effects on FXR at the concentration of 1.0 μM; while **5**, **6**, **7**, and **9** exhibited significant agonistic effects on FXR at the concentration of 10.0 μM. Moreover, **1** displayed a moderate significant agonistic effect on FXR at the concentration of 10.0 μM (Figure 12). Compound **10** exhibited a significant agonistic effect on PXR at the concentration of 10.0 nM, and even a higher agonistic effect on PXR as compared to that of the positive control, rifampicin, at the same concentration of 10.0 μM (Figure 13).

## 3. Materials and Methods

### 3.1. General Methods

Optical rotations were recorded at room temperature on a MCP200 modular circular polarimeter (Anton Paar GmbH, Seelze, Germany). A GENESYS 10S UV–Vis spectrophotometer (Thermo Scientific, Shanghai, China) was used to obtain UV spectra. The NMR spectroscopic data were measured on a Bruker AV-400 NMR spectrometer (Bruker Scientific Technology Co. Ltd., Karlsruhe, Germany) using TMS as the internal standard. Single-crystal X-ray diffraction analyses were carried out on an Agilent Xcalibur Atlas Gemini Ultra-diffractometer with mirror monochromated Cu Kα radiation (λ = 1.54184 Å) at 100 K. An LC-ESI (Bruker Daltonics, Bremen, Germany) and an LC-ESI-QTOF mass spectrometer (SYNAPTTM G2 HDMS, Waters, Manchester, UK) were used to acquire HR-ESIMS data. For electronic circular dichroism (ECD) spectra, a Jasco 810 spectropolarimeter (JASCO Corporation, Tokyo, Japan) was applied with the solvent of acetonitrile. Semi-preparative HPLC was carried out on a Waters 2535 pump equipped with a 2489 UV detector (Waters Corporation, Milford, NY, USA) and an ODS column (YMC, 250 × 10 mm inner diameter, 5 µm). Silica gel (100–200 mesh, Qingdao Mar. Chem. Ind. Co. Ltd., Qingdao, China) and ODS silica gel (A-HG 12 nm, 50 mm, YMC Co. Ltd., Kyoto, Japan) were used for column chromatography.

### 3.2. Plant Material

The seeds of two mangrove plants, *Xylocarpus granatum* and *Xylocarpus moluccensis*, were collected in September 2007 at the mangrove swamps of Krishna estuary, Andhra Pradesh, India. The plant species were identified by Mr. Tirumani Satyanandamurty (Government Degree College at Amadala Valasa, India). The voucher samples (No. IXG-02 for *X*. *granatum*) and (No. IXM200701 for *X*. *moluccensis*) were kept in the Marine Drugs Research Center, College of Pharmacy, Jinan University.

The stems and twigs of the semi-mangrove plant, *Excoecaria agallocha*, were collected in December 2003 at the mangrove swamps of Hainan Island, China. The identification of the plant was done by Professor Jun Wu (School of Pharmaceutical Sciences, Southern Medical University). A voucher sample (EA-J1) was kept in Marine Drugs Research Center, College of Pharmacy, Jinan University.

### 3.3. Extraction and Isolation

The seeds of *X. granatum* were air-dried (22.0 kg), powdered, and then extracted with 95% EtOH (5 × 75 L) at room temperature. The resulting EtOH extract (4.0 kg) was partitioned between EtOAc/water (3:1, *v*/*v*) to afford the EtOAc portion (1100.0 g). The EtOAc portion (250.0 g) was subjected to silica gel column chromatography (120 × 10 cm inner diameter; chloroform/methanol, from 100:0 to 10:1) to give 210 fractions. Fractions 50–82 (20.0 g) were combined and separated by an ODS silica gel column (100 × 7 cm inner diameter; acetone/water, from 40:60 to 100:0) to afford 81 subfractions. The subfraction 52 (308.0 mg) was purified by semi-preparative HPLC (MeCN/H_2_O, 40:60, 3.0 mL/min) to give **1** (78.0 mg, t*_R_* = 54.0 min) and **5** (3.0 mg, t*_R_* = 62.0 min); whereas the subfraction 53 (128.0 mg) was separated by semi-preparative HPLC (MeCN/H_2_O, 40:60, 3.0 mL/min) to yield **4** (15.5 mg, t*_R_* = 35.8 min).

The seeds of *X. moluccensis* were air-dried (6.0 kg), powdered, and extracted with 95% EtOH (6 *×* 15 L) at room temperature. The resulting extract (824.6 g) was partitioned between EtOAc/water (3:1, *v*/*v*) to give the EtOAc portion (299.1 g), which was applied to silica gel column chromatography (150 *×* 8.5 cm inner diameter; chloroform/methanol, from 100:0 to 5:1) to afford 223 fractions. Fractions 9–11 (18.5 g) were combined and further separated by ODS silica gel column chromatography (60 *×* 3 cm inner diameter; acetone/water, from 50:50 to 100:0) to afford 90 subfractions. The subfraction 61 (800.0 mg) was purified by semi-preparative HPLC (MeOH/H_2_O, 82:18, 3.0 mL/min) to yield **2** (6.5 mg, t*_R_* = 45.0 min) and **3** (1.0 mg, t*_R_* = 52.0 min). Fractions 68–71 (22.9 g) were combined and separated by ODS silica gel column chromatography (60 *×* 3 cm inner diameter.; acetone/water, from 40:60 to 100:0) to give 111 subfractions. The subfraction 75 (210.0 mg) was purified by semi-preparative HPLC (MeCN/H_2_O, 58:42, 3.0 mL/min) to afford **10** (0.9 mg, t*_R_* = 46.0 min).

The stems and twigs of *E.*
*agallocha* were air-dried (10.0 kg), powdered, and extracted with 95% EtOH (5 × 40.0 L) at room temperature. The resulting brown extract (370.0 g)was then suspended in water, and extracted with hexane (3:1, *v*/*v*) and EtOAc (3:1, *v*/*v*), successively, to give the EtOAc portion (192.0 g), which was separated by silica gel column chromatography (150 × 10.5 cm inner diameter; chloroform/methanol, from 100:0 to 5:1) to yield 285 fractions. Fractions 130 to 170 (17.5 g) were combined and subjected to ODS silica gel column chromatography (110 × 6 cm inner diameter; acetone/water, from 30:70 to 100:0) to give 76 subfractions. The subfraction 35 (1.3 g) was purified by semi-preparative HPLC (MeCN/H_2_O, 42:58, 3.0 mL/min) to yield **6** (4.0 mg, t*_R_* = 69.8 min) and **7** (4.0 mg, t*_R_* = 73.1 min).

Fractions 40 to 100 (18.0 g) were combined and further separated by ODS silica gel column chromatography (110 × 6 cm inner diameter; acetone/water, from 55:45 to 100:0) to yield 65 subfractions. The subfraction 33 (76.0 mg) was subjected to semi-preparative HPLC (MeCN/H_2_O, 70:30, 3.0 mL/min) to give **9** (3.3 mg, t*_R_* = 36.8 min). Fractions 101 to 129 (45.0 g) were combined and purified by ODS silica gel column chromatography (100 × 10 cm inner diameter; acetone/water, from 40:60 to 100:0) to yield 80 subfractions. The subfraction 44 (190.0 mg) was purified by semi-preparative HPLC (MeOH/H_2_O, 82:18, 3.0 mL/min) to afford **8** (4.0 mg, t*_R_* = 30.8 min).

Xylocarpol A (**1**): (5*R*,9*R*,10*R*,13*S*,14*S*,17*S*,20*R*,22*S*,23*S*,24*S*)-22,23,24,25-tetrahydroxytirucalla- 7-ene-3-one. Colorless crystals; ${\left[a\right]}_{D}^{25}$ = −68.0 (*c* 0.1, acetone); UV (MeCN) *λ*_max_ (log*ε*) 193 (4.20) nm; ^1^H- and ^13^C-NMR spectroscopic data see Table 1 and Table 2, respectively; HR-ESIMS *m*/*z* 525.3356 [M + Cl]^−^ (calculated for C_30_H_50_ClO_5_, 525.3352).

Xylocarpol B (**2**): (5*R*,9*R*,10*R*,13*S*,14*S*,17*S*,20*R*,22*S*)-22-hydroxytirucalla-7-ene-3,23-dione. Colorless crystals; ${\left[a\right]}_{D}^{25}$ = −8.0 (*c* 0.1, acetone); UV (MeCN) *λ*_max_ (log*ε*) 191 (4.16) nm; ^1^H- and ^13^C-NMR spectroscopic data see Table 1 and Table 2, respectively; HR-ESIMS *m*/*z* 457.3670 [M + H]^+^ (calculated for C_30_H_49_O_3_, 457.3676).

Xylocarpol C (**3**): White amorphous power; ${\left[a\right]}_{D}^{25}$ = −82.0 (*c* 0.1, acetone); UV (MeCN) *λ*_max_ (log*ε*) 193 (3.77) nm; ^1^H- and ^13^C-NMR spectroscopic data see Table 1 and Table 2, respectively; HR-ESIMS *m*/*z* 457.3654 [M + H]^+^ (calculated for C_30_H_49_O_3_, 457.3676).

Xylocarpol D (**4**): (3*S*,5*R*,9*R*,10*R*,13*S*,14*S*,17*S*,20*R*,22*S*,23*R*,24*S*)-3,23,24-trihydroxy-22,25- epoxytirucalla-7-ene. White amorphous power; ${\left[a\right]}_{D}^{25}$ = −70.0 (*c* 0.1, acetone); UV (MeCN) *λ*_max_ (log*ε*) 203 (4.40) nm; ^1^H- and ^13^C-NMR spectroscopic data see Table 1 and Table 2, respectively; HR-ESIMS *m*/*z* 509.3402 [M + Cl]^−^ (calculated for C_30_H_50_ClO_4_, 509.3403).

Xylocarpol E (**5**): White amorphous power; ${\left[a\right]}_{D}^{25}$ = −8.0 (*c* 0.1, acetone); UV (MeCN) *λ*_max_ (log*ε*) 194 (3.70) nm; ^1^H- and ^13^C-NMR spectroscopic data see Table 1 and Table 2, respectively; HR-ESIMS *m*/*z* 507.3245 [M + Cl]^−^ (calculated for C_30_H_48_ClO_4_, 507.3247).

Agallochol A (**6**): (3*S*,5*R*,9*R*,10*R*,13*S*,14*S*,17*S*,20*S*,24*S*)-3,24,25-trihydroxytirucalla-7-ene-6-one. White amorphous power; ${\left[a\right]}_{D}^{25}$ = −19.5 (*c* 0.1, acetone); UV (MeCN) *λ*_max_ (log*ε*) 204 (3.61), 246 (3.68) nm; ^1^H- and ^13^C-NMR spectroscopic data see Table 3 and Table 4, respectively; HR-ESIMS *m*/*z* 475.3790 [M + H]^+^ (calculated for C_30_H_51_O_4_, 475.3782).

Agallochol B (**7**): (3*S*,5*R*,9*R*,10*R*,13*S*,14*S*,17*S*,20*S*,24*R*)-3,24,25-trihydroxytirucalla-7-ene-6-one. White amorphous power; ${\left[a\right]}_{D}^{25}$ = −13.2 (*c* 0.1, acetone); UV (MeCN) *λ*_max_ (log*ε*) 204 (3.55), 246 (3.49) nm; ECD (0.21 mM, MeCN) λ_max_ (∆*ε*) 209.0 (−16.1), 242.6 (+ 12.8) nm; ^1^H- and ^13^C-NMR spectroscopic data see Table 3 and Table 4, respectively; HR-ESIMS *m*/*z* 475.3792 [M + H]^+^ (calculated for C_30_H_51_O_4_, 475.3782).

Agallochol C (**8**): (3*S*,5*R*,9*R*,10*R*,13*S*,14*S*,17*S*,20*S*)-3,25-dihydroxytirucalla-7,23-diene-6-one. White amorphous power; ${\left[a\right]}_{D}^{25}$ = −22.4 (*c* 0.1, acetone); UV (MeCN) *λ*_max_ (log*ε*) 202 (3.07), 245 (3.82) nm; ECD (0.37 mM, MeCN) λ_max_ (∆*ε*) 209.6 (−10.4), 242.4 (+ 7.5) nm; ^1^H- and ^13^C-NMR spectroscopic data see Table 3 and Table 4, respectively; HR-ESIMS *m*/*z* 457.3673 [M + H]^+^ (calculated for C_30_H_49_O_3_, 457.3682).

Agallochol D (**9**): White amorphous power; ${\left[a\right]}_{D}^{25}$= − 18.4 (*c* 0.1, acetone); UV (MeCN) *λ*_max_ (log*ε*) 204 (3.19) nm; ^1^H- and ^13^C-NMR spectroscopic data see Table 3 and Table 4, respectively; HR-ESIMS *m*/*z* 451.2985 [M + Cl]^+^ (calculated for C_27_H_44_ClO_3_, 451.2984).

25-dehydroxy protoxylogranatin B (**10**): White amorphous power; ${\left[a\right]}_{D}^{25}$ = −22.0 (*c* 0.09, acetone); UV (MeCN) *λ*_max_ (log*ε*) 202 (4.20) nm; ^1^H- and ^13^C-NMR spectroscopic data see Table 3 and Table 4, respectively; HR-ESIMS *m*/*z* 543.3303 [M + H]^−^ (calculated for C_32_H_47_O_7_, 543.3316).

### 3.4. Preparation of the (3,23,24)-Tri(S)- and (3,23,24)-Tri(R)-MTPA Esters of **4**

Compound **4** (1.0 mg) was treated with (*S*)-MTPA acid (2.0 mg), 4-dimethylaminopyridine (DMAP) (1.0 mg), and *N*,*N*′-dicyclohexylcarbodiimide (DCC) (4.0 mg) in dried dichloromethane (0.5 mL) at room temperature for 72 h. The reaction mixture was concentrated and purified by C_18_ reversed-phase HPLC (YMC-Pack 250 *×* 10 mm i.d.) with acetonitrile (MeCN/H_2_O, 100:0) to afford the (3,23,24)-tri(*S*)-MTPA esters of **4**, named **4s** (1.7 mg). Similarly, the (3,23,24)-tri(*R*)-MTPA esters of **4**, named **4r** (1.8 mg), were prepared in the same way.

**4s**: amorphous powder; ^1^H NMR (CDCl_3_, 400 MHz): *δ*_H_ 1.252 (1H, m, H-1*α*), 1.705 (1H, m, H-1*β*), 1.637 (1H, m, H-2*α*), 1.797 (1H, m, H-2*β*), 4.730 (1H, dd, *J* = 11.2, 3.2 Hz, H-3), 1.454 (1H, m, H-5), 0.795 (3H, s, H_3_-18), 0.752 (3H, s, H_3_-19), 1.647 (1H, m, H-20), 0.910 (3H, overlapped, H_3_-21), 3.853 (1H, d, *J* = 5.2 Hz, H-22), 5.365 (1H, t, *J* = 5.2 Hz, H-23), 5.000 (1H, d, *J* = 6.0 Hz, H-24), 1.055 (3H, s, H_3_-26), 1.288 (3H, s, H_3_-27), 0.891 (3H, overlapped, H_3_-28), 0.900 (3H, s, H_3_-29), 0.967 (3H, s, H_3_-30).

**4r**: amorphous powder; ^1^H NMR (CDCl_3_, 400 MHz): *δ*_H_ 1.264 (1H, m, H-1*α*), 1.736 (1H, m, H-1*β*), 1.751 (1H, m, H-2*α*), 1.868 (1H, m, H-2*β*), 4.757 (1H, dd, *J* = 11.6, 4.4 Hz, H-3), 1.442 (1H, m, H-5), 0.820 (3H, s, H_3_-18), 0.778 (3H, s, H_3_-19), 1.803 (1H, m, H-20), 0.929 (3H, d, *J* = 6.0 Hz, H_3_-21), 4.056 (1H, d, *J* = 4.8 Hz, H-22), 5.403 (1H, t, *J* = 5.2 Hz, H-23), 4.926 (1H, d, *J* = 6.4 Hz, H-24), 0.989 (3H, s, H_3_-26), 1.216 (3H, s, H_3_-27), 0.884 (3H, overlapped, H_3_-28), 0.806 (3H, s, H_3_-29), 0.968 (3H, s, H_3_-30).

### 3.5. Preparation of the (3,24)-Di(S)- and (3,24)-Di(R)-MTPA Esters of **6** and **7**

Compound **6** (1.0 mg) was treated with (*S*)-MTPA acid (2.0 mg), DMAP (1.0 mg), and DCC (4.0 mg) in dried dichloromethane (0.5 mL) at room temperature for 72 h. The reaction mixture was concentrated and purified by C_18_ reversed-phase HPLC (YMC-Pack 250 *×* 10 mm i.d.) with aqueous acetonitrile (MeCN/H_2_O, 84:16) to afford the (3,24)-di(*S*)-MTPA esters of **6**, named **6s** (0.5 mg). The (3,24)-di(*R*)-MTPA esters of **6**, named **6r** (0.7 mg), was prepared in the same way.

**6s**: amorphous powder; ^1^H NMR (CDCl_3_, 400 MHz): *δ*_H_ 1.497 (1H, m, H-1*α*), 1.717 (1H, m, H-1*β*), 1.619 (1H, m, H-2*α*), 1.780 (1H, m, H-2*β*), 4.672 (1H, dd, *J* = 11.2, 4.0 Hz, H-3), 2.220 (1H, s, H-5), 5.699 (1H, d, *J* = 2.8 Hz, H-7), 0.806 (3H, s, H_3_-18), 0.867 (3H, s, H_3_-19), 1.358 (1H, m, H-20), 0.784 (3H, d, *J* = 6.0 Hz, H_3_-21), 1.353 (1H, m, H-22a), 0.876 (1H, m, H-22b), 1.613 (1H, m, H-23a), 1.547 (1H, m, H-23b), 4.964 (1H, dd, *J =* 10.0, 2.4 Hz, H-24), 1.163 (3H, s, H_3_-26), 1.228 (3H, s, H_3_-27), 1.220 (3H, s, H_3_-28), 1.139 (3H, s, H_3_-29), 1.032 (3H, s, H_3_-30).

**6r**: amorphous powder; ^1^H NMR (CDCl_3_, 400 MHz): *δ*_H_ 1.520 (1H, m, H-1*α*), 1.762 (1H, m, H-1*β*), 1.749 (1H, m, H-2*α*), 1.850 (1H, m, H-2*β*), 4.703 (1H, dd, *J* = 11.6, 4.4 Hz, H-3), 2.222 (1H, s, H-5), 5.697 (1H, d, *J* = 2.4 Hz, H-7), 0.825 (3H, s, H_3_-18), 0.895 (3H, s, H_3_-19), 1.416 (1H, m, H-20), 0.854 (3H, d, *J* = 6.0 Hz, H_3_-21), 1.445 (1H, m, H-22a), 1.009 (1H, m, H-22b), 1.710 (1H, m, H-23a), 1.645 (1H, m, H-23b), 4.981 (1H, dd, *J =* 10.0, 2.4 Hz, H-24), 1.148 (3H, overlapped, H_3_-26), 1.180 (3H, s, H_3_-27), 1.148 (3H, overlapped, H_3_-28), 1.148 (3H, overlapped, H_3_-29), 1.035 (3H, s, H_3_-30).

Compound **7** (1.0 mg) was treated with (*S*)-MTPA acid (2.0 mg), DMAP (1.0 mg), and DCC (4.0 mg) in dried dichloromethane (0.5 mL) at room temperature for 72 h. The reaction mixture was concentrated and purified by C_18_ reversed-phase HPLC (YMC-Pack 250 *×* 10 mm i.d.) with aqueous acetonitrile (MeCN/H_2_O, 90:10) to afford (3,24)-di(*S*)-MTPA esters of **7**, named **7s** (1.5 mg). The (3,24)-di(*R*)-MTPA esters of **7**, named **7r** (1.4 mg), was prepared in the same way.

**7s**: amorphous powder; ^1^H NMR (CDCl_3_, 400 MHz): *δ*_H_ 1.505 (1H, m, H-1*α*), 1.722 (1H, m, H-1*β*), 1.625 (1H, m, H-2*α*), 1.787 (1H, m, H-2*β*), 4.673 (1H, dd, *J* = 11.6, 4.0 Hz, H-3), 2.220 (1H, s, H-5), 5.694 (1H, d, *J* = 2.4 Hz, H-7), 0.815 (3H, s, H_3_-18), 0.864 (3H, s, H_3_-19), 1.333 (1H, m, H-20), 0.890 (3H, d, *J* = 6.4 Hz, H_3_-21), 1.853 (1H, m, H-22a), 1.052 (1H, m, H-22b), 1.861 (1H, m, H-23a), 1.444 (1H, m, H-23b), 4.951 (1H, dd, *J =* 9.6, 2.4 Hz, H-24), 1.151 (3H, s, H_3_-26), 1.183 (3H, s, H_3_-27), 1.228 (3H, s, H_3_-28), 1.137 (3H, s, H_3_-29), 1.029 (3H, s, H_3_-30).

**7r**: amorphous powder; ^1^H NMR (CDCl_3_, 400 MHz): *δ*_H_ 1.523 (1H, m, H-1*α*), 1.747 (1H, m, H-1*β*), 1.747 (1H, m, H-2*α*), 1.859 (1H, m, H-2*β*), 4.699 (1H, dd, *J* = 11.2, 4.0 Hz, H-3), 2.214 (1H, s, H-5), 5.692 (1H, d, *J* = 2.8 Hz, H-7), 0.791 (3H, s, H_3_-18), 0.893 (3H, s, H_3_-19), 1.272 (1H, m, H-20), 0.845 (3H, d, *J* = 6.4 Hz, H_3_-21), 1.743 (1H, m, H-22a), 0.945 (1H, m, H-22b), 1.716 (1H, m, H-23a), 1.397 (1H, m, H-23b), 4.948 (1H, dd, *J =* 10.0, 2.4 Hz, H-24), 1.162 (3H, s, H_3_-26), 1.224 (3H, s, H_3_-27), 1.148 (3H, overlapped, H_3_-28), 1.145 (3H, overlapped, H_3_-29), 1.028 (3H, s, H_3_-30).

### 3.6. X-Ray Crystal Data for Xylocarpols A–B (**1**–**2**)

Xylocarpol A (**1**): orthorhombic, C_31_H_54_O_6_ (C_30_H_50_O_5_·CH_3_OH), space group *P*2_1_, a = 35.3234(5) Å, b = 11.41820(10) Å, c = 7.38890(10) Å, *α* = 90°, *β* = 90°, *γ* = 90°, V = 2980.16(6) Å^3^, Z = 4, D_calcd_ = 1.165 g cm^−3^, *μ* = 0.624 mm^−1^. Crystal size: 0.14 × 0.13 × 0.12 mm^3^, 33,526 measured reflections, 5916 (R_int_ = 0.0475) independent reflections, 351 parameters, 0 restraints, *F* (000) = 1152.0, *R*_1_ = 0.0429, *wR*_2_ = 0.1120 (all data), *R*_1_ = 0.0426, *wR*_2_ = 0.1118 (I > 2σ(I)), and goodness-of-fit (*F^2^*) = 1.030. The absolute structural parameter was 0.01 (6), the Flack x parameter was 0.01(6), and the Hooft y was 0.01(5).

Xylocarpol B (**2**): orthorhombic, C_30_H_48_O_3_, space group *P*2_1_, a = 6.61860 (11) Å, b = 18.2616 (3) Å, c = 21.8783 (3) Å, *α* = 90°, *β* = 90°, *γ* = 90°, V = 2644.36 (7) Å^3^, Z = 4, D_calcd_ = 1.147 g cm^−3^, *μ* = 0.551 mm^−1^, crystal size: 0.14 × 0.12 × 0.11 mm^3^, 13,817 measured reflections, 5179 (R_int_ = 0.0367) independent reflections, 307 parameters, 0 restraints, *F*(000) = 1008.0, *R*_1_ = 0.0467, *wR*_2_ = 0.1178 (all data), *R*_1_ = 0.0443, *wR*_2_ = 0.1147 (I > 2σ(I)), and goodness-of-fit (F^2^) = 1.025. The absolute structural parameter was 0.08 (13), the Flack x parameter was 0.09(14), and the Hooft y was 0.02(13).

CCDC-1872275 (**1**) and 1872277 (**2**) contained the supplementary crystallographic data for this paper (excluding structure factors). These data were provided free of charge by The Cambridge Crystallographic Data Centre.

### 3.7. FXR Activation Bioassay

By cloning genes encoding FXR into pCI-neomammalian expression vector, the human FXR expression plasmid was constructed. By cloning a genomic DNA fragment upstream of the transcription start site into the luciferase vector pGL4.14 (luc2/Hygro) based on the previous method [29], bile salt export pump (BSEP) promoter reporter was constructed. Human hepatoma HepG2 cells were transiently transfected with the expression plasmid of FXR (100.0 ng), the reporter vector of BSEP promoter luciferase (100.0 ng), and the null-Renilla luciferase plasmid (10.0 ng) as an internal control. After the incubation for 24 h, cells were treated with vehicle DMSO (0.1%), compounds **1**, and **4**–**9**, respectively, at different concentrations (10.0, 100.0 nM, 1.0, or 10.0 µM) for 24 h. Then, cells were harvested for the determination of luciferase activity. Chenodeoxycholic acid (CDCA) was used as the positive control at the final concentration of 80.0 μM.

### 3.8. PXR Activation Bioassay

As described previously [30], the expression vector of human PXR and the human PXR XREM-driven luciferase reporter plasmid (CYP3A4XREM-luciferase) were constructed. By using the lipofectamine 3000 (Invitrogen, Carlsbad, CA., USA), human hepatoma HepG2 cells were transfected with expression and reporter plasmids, along with pGL4.74 (Promega, Beijing, China) as an internal standard. Then, cells were treated with vehicle DMSO (0.1%), compounds **1**, and **4**–**10**, respectively, at different concentrations (10.0, 100.0 nM, 1.0, or 10.0 µM) for 24 h. By using the Dual-luciferase^®^ Reporter Assay System (Promega, Beijing, China), the luciferase activities of the above compounds were recorded. By dividing the Firefly luciferase signal by the Renilla luciferase signal, the co-transfected plasmid results were normalized. Rifampicin, the well-known human PXR agonist, was used as the positive control at the final concentration of 10.0 μM.

## 4. Conclusions

In summary, 10 new triterpenoid compounds, including nine tirucallanes (**1**–**9**) and one apotirucallane (**10**), were obtained from mangrove plants *X**. granatum*, *X**. moluccensis*, and *E**. agallocha.* The absolute configurations of **1**, **2**, **4**, and **6**–**8** were unequivocally determined by single-crystal X-ray diffraction analysis (Cu K*α*), modified Mosher’s method, and the comparison of experimental ECD spectra. Compounds **5**, **6**, **7**, and **9** displayed potent activation effects on FXR at the concentration of 10.0 μM; whereas **10** exhibited very significant agonistic effects on PXR at nanomolar concentration. This work demonstrated that mangrove tirucallane- and apotirucallane-type triterpenoid compounds are valuable natural products for the discovery of leading compounds with potent activation effects on human FXR and PXR.

## Figures and Tables

**Figure 1 marinedrugs-16-00488-f001:**
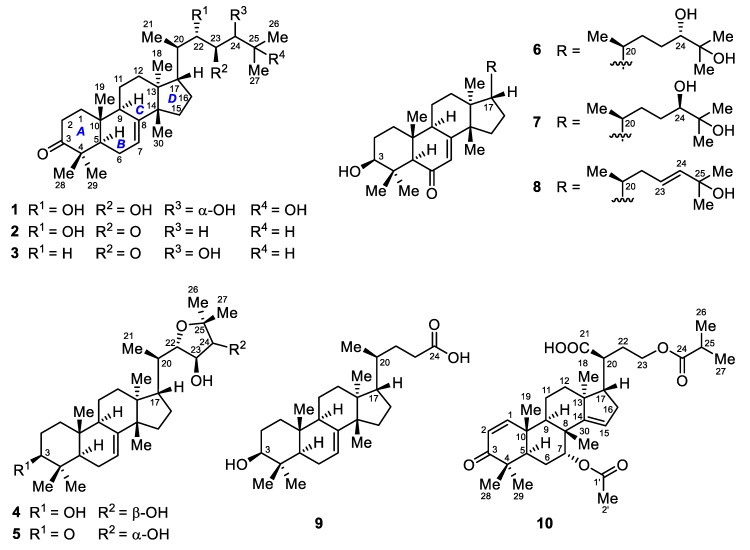
Structures of compounds **1**–**10**.

**Figure 2 marinedrugs-16-00488-f002:**
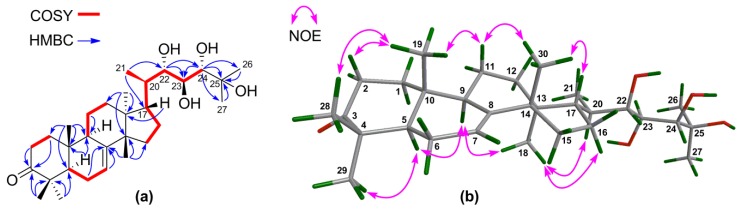
(**a**) Key ^1^H‒^1^H COSY and HMBC correlations for **1**; (**b**) Selected NOE interactions for **1**.

**Figure 3 marinedrugs-16-00488-f003:**
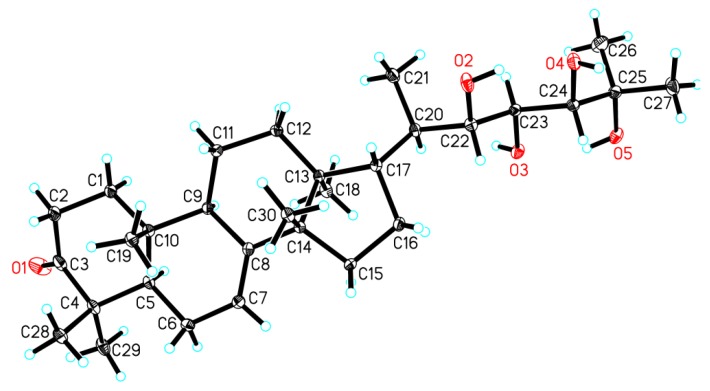
Perspective drawing of the single-crystal X-ray structure of **1**.

**Figure 4 marinedrugs-16-00488-f004:**
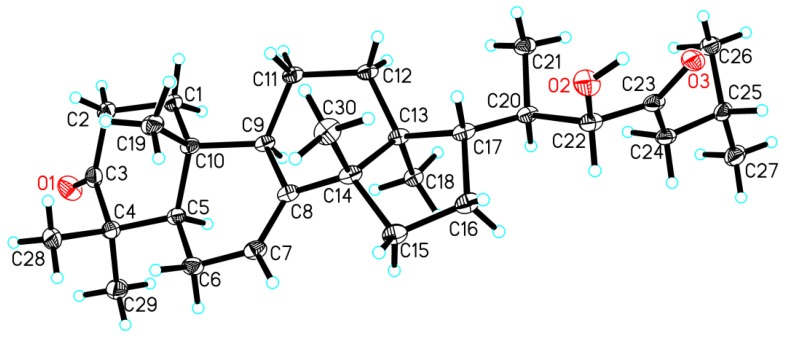
Perspective drawing of the single-crystal X-ray structure of **2**.

**Figure 5 marinedrugs-16-00488-f005:**
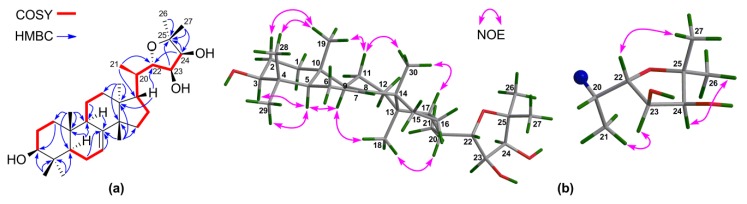
(**a**) Key ^1^H‒^1^H COSY and HMBC correlations for **4**; (**b**) Selected NOE interactions for **4**.

**Figure 6 marinedrugs-16-00488-f006:**
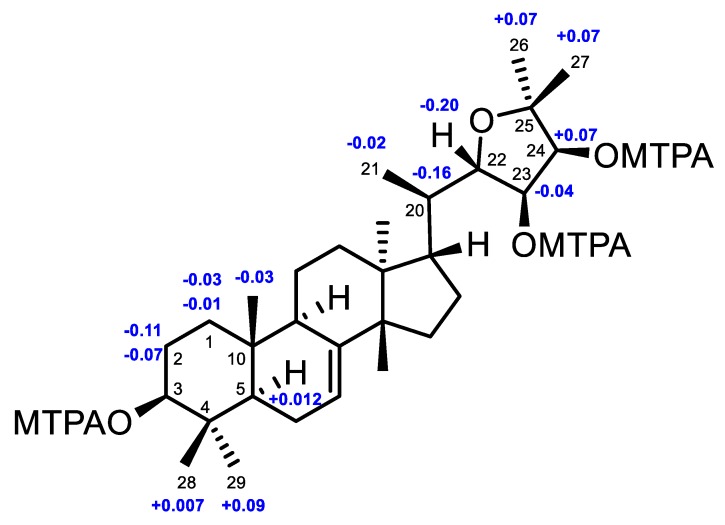
Δ*δ* values (Δ*δ* [ppm] = [*δ_S_* − *δ_R_*]) that were obtained for the (3,23,24)-tri(*S*)- and (3,23,24)-tri(*R*)-MTPA esters of **4** (**4s**, **4r**).

**Figure 7 marinedrugs-16-00488-f007:**
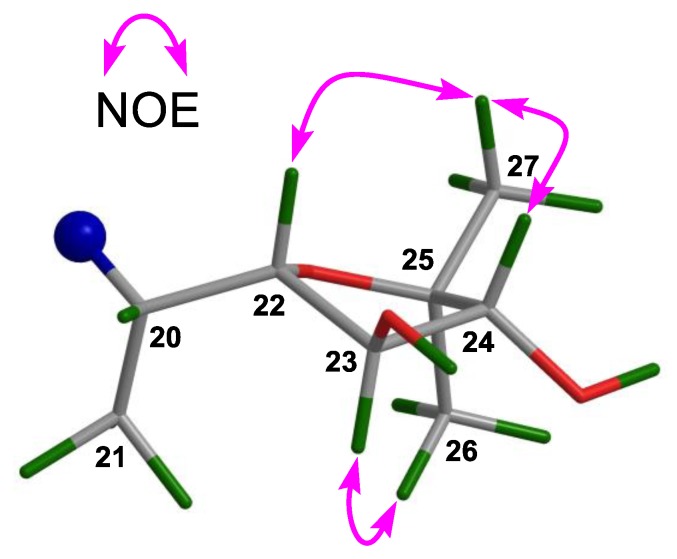
Diagnostic NOE interactions for the 2,2-dimethyltetrahydrofuran-3,4-diol moiety of **5**.

**Figure 8 marinedrugs-16-00488-f008:**
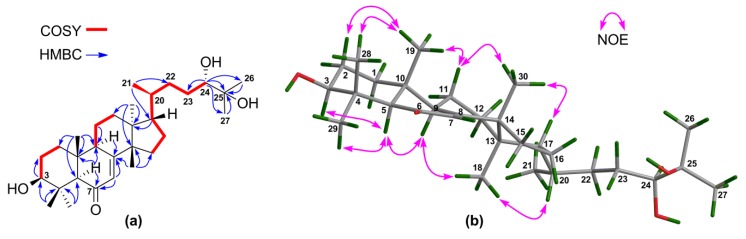
(**a**) Key ^1^H‒^1^H COSY and HMBC correlations for **6**; (**b**) Selected NOE interactions for **6**.

**Figure 9 marinedrugs-16-00488-f009:**
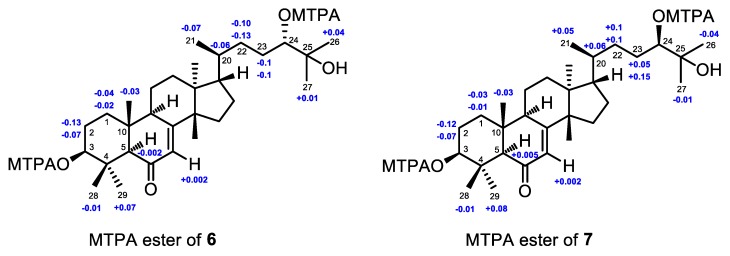
Δ*δ* values (Δ*δ* [ppm] = [*δ_S_* − *δ_R_*]) that were obtained for the (3,24)-di(*S*)- and (3,24)-di(*R*)-MTPA esters of **6** (**6s**, **6r**) and **7** (**7s**, **7r**).

**Figure 10 marinedrugs-16-00488-f010:**
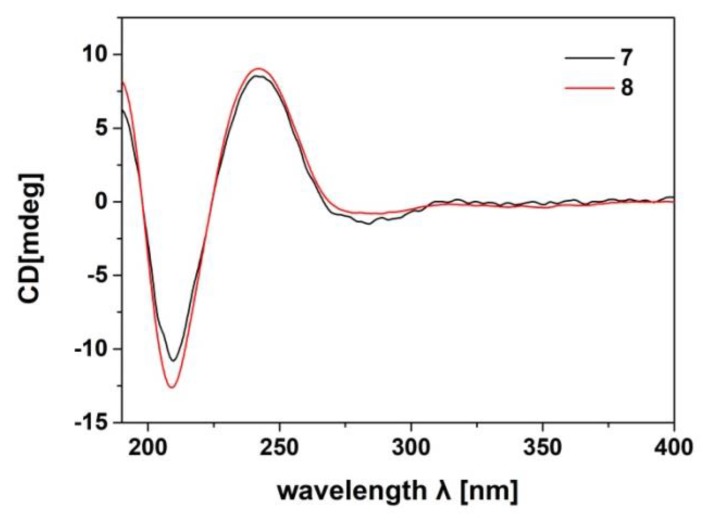
Comparison of the experimental ECD spectra of **7** and **8** (measured at concentrations of 100 and 167 µg/mL in acetonitrile, respectively).

**Figure 11 marinedrugs-16-00488-f011:**
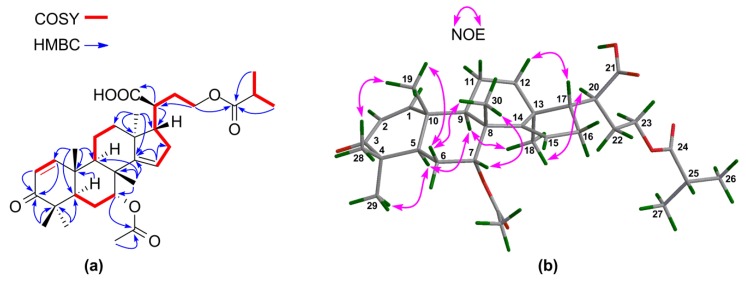
(**a**) Key ^1^H‒^1^H COSY and HMBC correlations for **10**; (**b**) Selected NOE interactions for **10**.

**Figure 12 marinedrugs-16-00488-f012:**
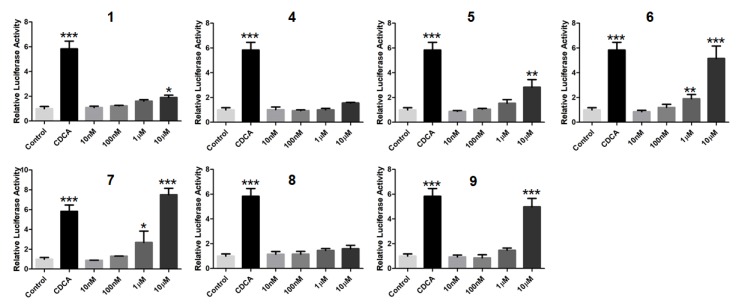
Agonistic effects of **1**, **4**–**9** on the farnesoid–X–receptor (FXR) in HepG2 cells. CDCA (80.0 µM) was used as the positive control. * *p* < 0.01, ** *p* < 0.001, *** *p* < 0.0001 compared to the control group.

**Figure 13 marinedrugs-16-00488-f013:**
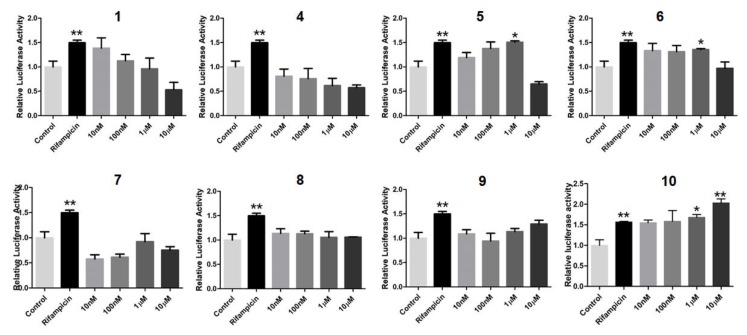
Agonistic effects of **1**, **4**–**10** on the pregnane–X–receptor (PXR) to modulate PXR target gene CYP3A4 transactivation in HepG2 cells. Rifampicin (10.0 µM) was used as the positive control. * *p* < 0.01, ** *p* < 0.001, *** *p* < 0.0001 compared to the control group.

**Table 1 marinedrugs-16-00488-t001:** ^1^H (400 MHz) NMR spectroscopic data for compounds **1**–**5** in CDCl_3_ (*δ* in ppm, *J* in Hz).

No.	1	2	3	4	5
1*α*	1.46 m	1.46 m	1.47 m	1.13 m	1.46 m
1*β*	1.99 m	2.00 m	2.00 m	1.67 ^a^	1.99 m
2*α*	2.25 dt (14.4, 3.2)	2.25 m	2.25 m	1.55 m	2.27 m
2*β*	2.76 td (14.4, 5.8)	2.77 td (14.8, 5.6)	2.75 td (14.4, 5.4)	1.65 ^a^	2.76 td (14.4, 3.6)
3				3.24 dd (11.2, 4.4)	
5	1.73 t (8.4)	1.74 t (8.4)	1.72 t (8.6)	1.30 m	1.72 t (8.4)
6*α*	2.10 ^a^	2.11 ^a^	2.11 ^a^	2.11 br s	2.10 ^a^
6*β*	2.10 ^a^	2.11 ^a^	2.11 ^a^	1.94 m	2.10 ^a^
7	5.32 d (2.8)	5.34 br d (3.2)	5.31 br d (2.8)	5.26 br s	5.32 d (3.2)
9	2.30 m	2.29 ^a^	2.29 ^a^	2.19 m	2.28 m
11*α*	1.63	1.57 ^a^	1.58 ^a^	1.55 m	1.55 m
11*β*	1.54 ^a^	1.57 ^a^	1.58 ^a^	1.48 ^a^	1.60 m
12*α*	1.66 m	1.57 ^a^	1.67 m	1.58 m	1.62 m
12*β*	1.83 m	1.84 m	1.81 m	1.83 m	1.84 m
15*α*	1.53 ^a^	1.56 ^a^	1.52 ^a^	1.48 ^a^	1.51 ^a^
15*β*	1.53 ^a^	1.56 ^a^	1.52 ^a^	1.48 ^a^	1.51 ^a^
16*α*	1.40 m	1.42 m	1.24 m	1.34 m	1.32 m
16*β*	2.00 m	2.14 m	1.92 m	2.00 m	2.00 m
17	1.81 m	2.01 m	1.56 m	1.84 m	1.92 m
18	0.87 s	0.87 s	0.86 s	0.82 s	0.82 s
19	1.01 s	1.02 s	1.01 s	0.74 s	1.01 s
20	1.97 m	1.87 m	2.05 m	1.64 m	1.60 m
21	0.94 d (6.8)	0.67 d (6.4)	0.91 d (6.4)	0.83 ^a^	0.90 d (6.8)
22a	3.80 br d (8.4)	4.14 s	2.51 br d (14.4)	3.81 d (6.0)	3.74 d (4.0)
22b			2.29 ^a^		
23	3.66 dd (8.4, 6.4)			3.96 t (6.0)	4.25 t (5.2)
24a	3.55 br d (6.4)	2.32 ^a^	4.07 s	3.65 d (6.0)	4.35 d (5.6)
24b		2.32 ^a^			
25		2.21 m	2.16 m		
26	1.35 ^a^	0.93 d (6.4)	0.71 d (6.4)	1.21 ^a^	1.42 s
27	1.35 ^a^	0.96 d (6.4)	1.13 d (6.4)	1.21 ^a^	1.36 s
28	1.12 s	1.13 s	1.12 s	0.85 s	1.12 s
29	1.05 ^a^	1.06 s	1.05 s	0.96 s	1.05 ^a^
30	1.05 ^a^	1.07 s	1.02 s	1.00 s	1.05 ^a^

^a^ Overlapped signals assigned by ^1^H–^1^H COSY, HSQC, and HMBC spectra without designating multiplicity.

**Table 2 marinedrugs-16-00488-t002:** ^13^C (100 MHz) NMR spectroscopic data for compounds **1**‒**5** in CDCl_3_ (*δ* in ppm).

No.	1	2	3	4	5
1	38.5 CH_2_	38.5 CH_2_	38.5 CH_2_	37.2 CH_2_	38.5 CH_2_
2	34.9 CH_2_	34.9 CH_2_	34.9 CH_2_	27.8 CH_2_	34.9 CH_2_
3	217.0 qC	217.0 qC	216.9 qC	79.3 CH	217.2 qC
4	47.9 qC	47.9 qC	47.9 qC	39.0 qC	47.9 qC
5	52.3 CH	52.3 CH	52.3 CH	50.6 CH	52.3 CH
6	24.4 CH_2_	24.4 CH_2_	24.4 CH_2_	23.9 CH_2_	24.4 CH_2_
7	118.0 CH	118.0 CH	118.1 CH	117.8 CH	117.9 CH
8	145.7 qC	145.7 qC	145.6 qC	145.8 qC	145.8 qC
9	48.5 CH	48.4 CH	48.4 CH	49.0 CH	48.5 CH
10	35.0 qC	35.0 qC	35.0 qC	34.9 qC	35.0 qC
11	18.3 CH_2_	18.2 CH_2_	18.2 CH_2_	18.1 CH_2_	18.3 CH_2_
12	33.7 CH_2_	33.4 CH_2_	33.5 CH_2_	33.6 CH_2_	33.7 CH_2_
13	43.6 qC	43.3 qC	43.6 qC	43.6 qC	43.3 qC
14	51.3 qC	51.4 qC	51.3 qC	51.2 qC	51.3 qC
15	34.0 CH_2_	34.0 CH_2_	34.0 CH_2_	34.0 CH_2_	34.0 CH_2_
16	27.4 CH_2_	28.0 CH_2_	28.4 CH_2_	27.7 CH_2_	27.7 CH_2_
17	48.6 CH	48.9 CH	53.0 CH	49.2 CH	48.9 CH
18	22.0 CH_3_	21.9 CH_3_	22.0 CH_3_	21.8 CH_3_	21.9 CH_3_
19	12.8 CH_3_	12.8 CH_3_	12.8 CH_3_	13.1 CH_3_	12.8 CH_3_
20	36.3 CH	39.3 CH	32.7 CH	37.6 CH	37.4 CH
21	11.5 CH_3_	12.1 CH_3_	19.7 CH_3_	12.4 CH_3_	12.0 CH_3_
22	75.8 CH	79.5 CH	45.7 CH_2_	83.7 CH	74.2 CH
23	70.9 CH	212.3 qC	212.2 qC	72.9 CH	86.2 CH
24	80.1 CH	46.8 CH_2_	80.1 CH	77.3 CH	73.1 CH
25	74.1 qC	24.5 CH	31.3 CH	80.9 qC	86.0 qC
26	24.6 CH_3_	22.6 CH_3_	14.6 CH_3_	27.7 CH_3_	28.1 CH_3_
27	27.7 CH_3_	22.7 CH_3_	20.2 CH_3_	21.3 CH_3_	21.6 CH_3_
28	21.6 CH_3_	21.6 CH_3_	21.6 CH_3_	14.7 CH_3_	21.6 CH_3_
29	24.8 CH_3_	24.5 CH_3_	24.6 CH_3_	27.5 CH_3_	24.5 CH_3_
30	27.6 CH_3_	27.6 CH_3_	27.4 CH_3_	27.6 CH_3_	27.6 CH_3_

**Table 3 marinedrugs-16-00488-t003:** ^1^H (400 MHz) NMR spectroscopic data for compounds **6**‒**10** in CDCl_3_ (*δ* in ppm, *J* in Hz).

No.	6	7	8	9	10
1*α*	1.41 m	1.41 m	1.41 m	1.14 m	7.14 d (10.4)
1*β*	1.71 m	1.71 m	1.71 m	1.68 m	
2*α*	1.59 ^a^	1.59 ^a^	1.59 ^a^	1.54 m	5.87 d (10.0)
2*β*	1.68 m	1.67 m	1.68 m	1.66 m	
3	3.22 dd (11.6, 4.0)	3.22 dd (11.6, 4.0)	3.22 dd (11.6, 4.4)	3.25 dd (11.2, 4.0)	
5	2.13 s	2.13 s	2.13 s	1.32 m	2.17 m
6*α*				2.15 m	1.77 m
6*β*				1.95 m	1.92 ^a^
7	5.70 d (2.4)	5.70 d (2.8)	5.70 d (2.8)	5.26 d (3.6)	5.23 br s
9	2.72 m	2.72 m	2.71 m	2.20 m	2.20 m
11*α*	1.75 ^a^	1.75 ^a^	1.75 ^a^	0.96 ^a^	1.92 ^a^
11*β*	1.59 ^a^	1.58 ^a^	1.57 ^a^	1.50 ^a^	1.70 m
12*α*	1.74 ^a^	1.55 ^a^	1.74 ^a^	1.63 m	1.91 ^a^
12*β*	1.87 m	1.88 m	1.86 m	1.78 m	1.56 m
15*α*	1.55 ^a^	1.55 ^a^	1.59 ^a^	1.52 ^a^	5.30 s
15*β*	1.74 ^a^	1.75 ^a^	1.54 ^a^	1.47 ^a^	
16*α*	1.38 m	1.38 m	1.37 m	1.31 m	2.04 m
16*β*	2.04 m	2.02 m	2.02 m	1.97 m	2.21 m
17	1.55 ^a^	1.55 ^a^	1.55 ^a^	1.47 ^a^	1.81 m
18	0.84 s	0.84 s	0.84 s	0.81 s	1.08 s
19	0.86 s	0.86 s	0.86 s	0.75 s	1.17 s
20	1.43 m	1.44 m	1.48 m	1.44 m	2.42 ^a^
21	0.91 d (6.0)	0.91 d (6.4)	0.88 d (6.4)	0.89 d (6.0)	
22a	1.54 ^a^	1.78 m	2.21 m	1.83 m	2.47 m
22b	1.28 m	1.02 m	1.76 ^a^	1.32 m	2.41 ^a^
23a	1.39 m	1.59 m	5.60 dd (15.6, 4.8)	2.41 m	4.32 d (10.0)
23b	1.39 m	1.18 m		2.28 m	3.95 dd (11.2, 5.6)
24	3.36 br s	3.30 dd (11.6, 1.6)	5.62 d (15.6)		
25					2.56 m
26	1.18 s	1.17 s	1.32 ^a^		1.17 s
27	1.23 s	1.23 s	1.32 ^a^		1.19 s
28	1.13 s	1.13 s	1.13 s	0.86 s	1.08 s
29	1.32 s	1.32 s	1 32 ^a^	0.97 ^a^	1.08 s
30	1.06 s	1.05 s	1.04 s	0.97 ^a^	1.19 s
2′					1.95 s

^a^ Overlapped signals assigned by ^1^H–^1^H COSY, HSQC, and HMBC spectra without designating multiplicity.

**Table 4 marinedrugs-16-00488-t004:** ^13^C (100 MHz) NMR spectroscopic data for compounds **6**‒**10** in CDCl_3_ (*δ* in ppm).

No.	6	7	8	9	10
1	37.0 CH_2_	37.0 CH_2_	37.0 CH_2_	37.2 CH_2_	158.1 CH
2	26.6 CH_2_	26.6 CH_2_	26.6 CH_2_	27.7 CH_2_	125.6 CH
3	79.1 CH	79.1 CH	79.1 CH	79.3 CH	204.6 qC
4	38.0 qC	38.0 qC	38.0 qC	39.0 qC	44.2 qC
5	65.2 CH	65.2 CH	65.2 CH	50.6 CH	46.1 CH
6	200.0 qC	200.0 qC	200.0 qC	23.9 CH_2_	23.8 CH_2_
7	125.0 CH	125.0 CH	125.0 CH	118.0 CH	74.6 CH
8	170.6 qC	170.6 qC	170.5 qC	145.7 qC	42.8 qC
9	50.4 CH	50.4 CH	50.4 CH	48.9 CH	38.4 CH
10	43.8 qC	43.8 qC	43.8 qC	34.9 qC	39.8 qC
11	17.6 CH_2_	17.6 CH_2_	17.5 CH_2_	18.1 CH_2_	16.7 CH_2_
12	32.9 CH_2_	32.8 CH_2_	32.6 CH_2_	33.8 CH_2_	34.1 CH_2_
13	43.0 qC	43.0 qC	43.0 qC	43.6 qC	46.5 qC
14	52.3 qC	52.3 qC	52.3 qC	51.1 qC	159.1 qC
15	32.8 CH_2_	32.9 CH_2_	32.9 CH_2_	34.0 CH_2_	118.9 CH
16	27.8 CH_2_	27.7 CH_2_	27.6 CH_2_	28.1 CH_2_	34.6 CH_2_
17	52.7 CH	52.6 CH	52.1 CH	52.8 CH	54.6 CH
18	21.9 CH_3_	21.9 CH_3_	21.9 CH_3_	21.9 CH_3_	19.7 CH_3_
19	14.3 CH_3_	14.3 CH_3_	14.3 CH_3_	13.1 CH_3_	19.0 CH_3_
20	35.8 CH	36.3 CH	36.2 CH	35.7 CH	35.6 CH
21	18.2 CH_3_	18.5 CH_3_	18.4 CH_3_	18.0 CH_3_	176.3 qC
22	32.8 CH_2_	33.2 CH_2_	38.6 CH_2_	30.9 CH_2_	35.1 CH_2_
23	28.4 CH_2_	28.7 CH_2_	125.2 CH	31.0 CH_2_	65.4 CH_2_
24	78.6 CH	79.5 CH	139.7 CH	178.5 qC	177.1 qC
25	73.2 qC	73.2 qC	70.8 qC		34.1 CH
26	23.3 CH_3_	23.2 CH_3_	30.0 CH_3_		19.0 CH_3_
27	26.7 CH_3_	26.6 CH_3_	29.9 CH_3_		19.1 CH_3_
28	14.8 CH_3_	14.8 CH_3_	14.8 CH_3_	14.7 CH_3_	21.3 CH_3_
29	28.3 CH_3_	28.3 CH_3_	28.3 CH_3_	27.6 CH_3_	27.1 CH_3_
30	24.9 CH_3_	24.9 CH_3_	24.9 CH_3_	27.3 CH_3_	27.4 CH_3_
1′					170.1 qC
2′					21.2 CH_3_

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
