# Peer review of "Mangrove Tirucallane- and Apotirucallane-Type Triterpenoids: Structure Diversity of the C-17 Side-Chain and Natural Agonists of Human Farnesoid/Pregnane–X–Receptor"

_marinedrugs, 2018, doi:10.3390/md16120488_

Reviewer 1 Report

With the manuscript 394744, the authors describe the structure elucidation of a series of previously undescribed triterpenoids isolated from three mangroves, as well as their effects towards FXR and PXR. The content is relevant i.e. the novelty of the compounds, thus being fully aligned with the scope of Marine Drugs. While clearly meriting its acceptance for publication, some minor changes and suggestions should be considered as mentioned below.

Firstly, while generally well written the quality of the writing should be improved. Just to cite a few, minor style and grammar mistakes include:

Line 22: “…were isolated from the mangrove plants of Xylocarpus granatum, Xylocarpus moluccensis, …”.

Line 36: “Long-term cholestasis could can lead to primary biliary, …”

Line 44: “…have been used as the in folk medicine, particularly in countries of South Asia and Southeast Asian countries for the treatment of cholera,…”

Lines 48-50: “…led to the isolation and identification of various limonoids, protolimonoids, alkaloids, and flavanones, among which limonoids being identified are as the main secondary metabolites of these mangroves.”.

Lines 51-52: “…has been used in Asia as the a traditional medicine for the treatment of epilepsy…”.

The structural assignments of the compounds are consistent with the NMR data and their discussion is detailed enough. However, the authors should revise minor inaccuracies such as in:

Table 1. The multiplicity of the H-6α and H-6β were not defined (presumably due to overlapped signals).   

Line 125: Correct to “However, due to the limited amount of 3, the chirality of C-24 could not be determined.”

Lines 208-209: Since it is apparent that the authors were able to determine the coupling constants of H-23, it is unclear why it was not included on Table 3, where the splitting pattern was indicated as a multiplet.     

Table 4: Concerning compound 6, C-21 refers to a methyl group and not a methine.

Line 255: Correct H3-19/H-28 to H3-19/H3-28.

While providing the NMR data as supplementary material, it is my personal opinion (i.e. not mandatory) that it would be extremely useful to present data on the main COSY, HMBC and NOESY correlations in a table for a future and quick comparison by other researchers. Furthermore, despite the crystal-clear discussion on the elucidation of the compounds based on the NMR and HRMS data, thus not requiring further spectroscopic data, I am unaware if it is mandatory to include IR data as well.

The main downside of the current manuscript refers to the data concerning the effects upon FXR and PXR, lacking statistical analysis. In fact, while referring that compounds 5, 6, 7 and 9 led to a potent activation of FXR, it seems evident that 7 displayed a significantly stronger effect in comparison with CDCA. As such, the authors are advised to determine the levels of statistical significance. Similarly, it is scientifically inappropriate to refer that compound 10 exhibited very significant agonistic effects on PXR (Line 265), without determining the levels of significance through a proper statistical analysis.  

I am confident the current research article will be well received and will make an important contribution. That said, I do recommend acceptance of the current manuscript after minor revision, subject to tidying up some the highlighted issues indicated above.

Author Response

Response to Reviewer(s)' Comments

Reviewer: 1

Comments:

With the manuscript 394744, the authors describe the structure elucidation of a series of previously undescribed triterpenoids isolated from three mangroves, as well as their effects towards FXR and PXR. The content is relevant i.e. the novelty of the compounds, thus being fully aligned with the scope of Marine Drugs. While clearly meriting its acceptance for publication, some minor changes and suggestions should be considered as mentioned below.

Firstly, while generally well written the quality of the writing should be improved. Just to cite a few, minor style and grammar mistakes include:

Line 22: “…were isolated from the mangrove plants of Xylocarpus granatumXylocarpus moluccensis, …”.

Response: We made the changes in our manuscript according to the above suggestion.

Line 36: “Long-term cholestasis could can lead to primary biliary, …”

Response: We made the changes in our manuscript according to the above suggestion.

Line 44: “…have been used as the in folk medicine, particularly in countries of South Asia and Southeast Asiancountries for the treatment of cholera,…”

Response: We made the changes in our manuscript according to the above suggestion.

Lines 48-50: “…led to the isolation and identification of various limonoids, protolimonoids, alkaloids, and flavanones, among which limonoids being identified are as the main secondary metabolites of these mangroves.”.

Response: We made the changes in our manuscript according to the above suggestion.

Lines 51-52: “…has been used in Asia as the a traditional medicine for the treatment of epilepsy…”.

Response: We made the changes in our manuscript according to the above suggestion.

 The structural assignments of the compounds are consistent with the NMR data and their discussion is detailed enough. However, the authors should revise minor inaccuracies such as in:

Table 1. The multiplicity of the H-6α and H-6β were not defined (presumably due to overlapped signals).   

Response: Yes, we cannot define due to overlapped signals. We also add the sign “a” (means overlapped signals).

Line 125: Correct to “However, due to the limited amount of 3, the chirality of C-24 could not be determined.”

Response: We made the changes in our manuscript according to the above suggestion.

Lines 208-209: Since it is apparent that the authors were able to determine the coupling constants of H-23, it is unclear why it was not included on Table 3, where the splitting pattern was indicated as a multiplet.

Response: We made the changes in our manuscript according to the above suggestion. The coupling constants of H-23a were added in Table 3.

Table 4: Concerning compound 6, C-21 refers to a methyl group and not a methine.

 Response: We made the changes in our manuscript according to the above suggestion.

Line 255: Correct H3-19/H-28 to H3-19/H3-28.

Response: We made the changes in our manuscript according to the above suggestion.

While providing the NMR data as supplementary material, it is my personal opinion (i.e. not mandatory) that it would be extremely useful to present data on the main COSY, HMBC and NOESY correlations in a table for a future and quick comparison by other researchers. Furthermore, despite the crystal-clear discussion on the elucidation of the compounds based on the NMR and HRMS data, thus not requiring further spectroscopic data, I am unaware if it is mandatory to include IR data as well.

Response: Too many 1H‒1H COSY, HMBC and NOESY correlations can be observed for ten triterpenoid compounds. It is extremely hard work for us. Not all the obtained compounds have enough quantity to measure their IR data. 

The main downside of the current manuscript refers to the data concerning the effects upon FXR and PXR, lacking statistical analysis. In fact, while referring that compounds 567 and 9 led to a potent activation of FXR, it seems evident that 7 displayed a significantly stronger effect in comparison with CDCA. As such, the authors are advised to determine the levels of statistical significance. Similarly, it is scientifically inappropriate to refer that compound 10 exhibited very significant agonistic effects on PXR (Line 265), without determining the levels of significance through a proper statistical analysis.

Response: We appreciate the careful reading of our manuscript and valuable comments. We applied the statistical analysis to the agonistic effects of the compounds mentioned in our manuscript upon FXR and PXR. Considering the statistical results, compounds 6 and 7 showed significant agonistic effects on FXR at the concentration of 1.0 μM; while compounds 5, 6, 7, and 9 showed significant agonistic effects on FXR at the concentration of 10.0 μM. In addition, compound 1 also showed a moderate significant agonistic effect on FXR at the concentration of 10.0 μM. Compound 10 exhibited a significant agonistic effect on PXR at the concentration of 10.0 nM, and even a higher agonistic effect as compared to that of the positive control, rifampicin, at the same concentration of 10.0 μM. 

I am confident the current research article will be well received and will make an important contribution. That said, I do recommend acceptance of the current manuscript after minor revision, subject to tidying up some the highlighted issues indicated above.

Reviewer 2 Report

A very well written paper with sound experimental design. The triterpenoids have been extracted and isolated from a huge amount of plant material. Solution structures, including stereochemical assignments, have been obtained by a good  combination of NMR and CD experiments. X-ray structures have been deposited at Cambridge CDC Database.  One compound, in particular, resulted endowed with a high agonistic effect on pregnane X receptor. I would certainly recommend this paper for publication.

Author Response

Reviewer: 2

Comments:

A very well written paper with sound experimental design. The triterpenoids have been extracted and isolated from a huge amount of plant material. Solution structures, including stereochemical assignments, have been obtained by a good combination of NMR and CD experiments. X-ray structures have been deposited at Cambridge CDC Database. One compound, in particular, resulted endowed with a high agonistic effect on pregnane X receptor. I would certainly recommend this paper for publication.

Response: We improved our manuscript according to the above suggestion.